# Targeting RHAMM in Cancer: Crosstalk with Non-Coding RNAs and Emerging Therapeutic Strategies Including Peptides, Oligomers, Antibodies, and Vaccines

**DOI:** 10.3390/ijms26157198

**Published:** 2025-07-25

**Authors:** Dong Oh Moon

**Affiliations:** Department of Biology Education, Daegu University, 201, Daegudae-ro, Gyeongsan-si 38453, Gyeongsangbuk-do, Republic of Korea; domoon@daegu.ac.kr; Tel./Fax: +82-53-852-6992

**Keywords:** RHAMM, non-coding RNAs, peptide inhibitors, HA oligomers, antibodies, peptide vaccines

## Abstract

Cancer remains a major cause of mortality worldwide, driven by complex molecular mechanisms that promote metastasis and resistance to therapy. Receptor for hyaluronan-mediated motility (RHAMM) has emerged as a multifunctional regulator in cancer, contributing to cell motility, invasion, proliferation, and fibrosis. In addition to being regulated by non-coding RNAs (ncRNAs), including miRNAs, lncRNAs, and circRNAs, RHAMM serves as a promising therapeutic target. Recent developments in RHAMM-targeted strategies include function-blocking peptides (e.g., NPI-110, NPI-106, and P15-1), hyaluronan (HA) oligomers, and anti-RHAMM antibodies, all shown to modulate tumor stroma and suppress tumor invasiveness. Importantly, RHAMM-targeted peptide vaccines, such as the RHAMM-R3 epitope, have demonstrated immunogenicity and anti-leukemia efficacy in both pre-clinical and early clinical studies, suggesting their potential to elicit specific CD8^+^ T-cell responses and enhance graft-versus-leukemia effects. This review summarizes the intricate roles of RHAMM in cancer progression, its modulation by ncRNAs, and the translational promise of novel RHAMM-targeting approaches, providing insights into future directions for precision cancer therapy.

## 1. Introduction

Cancer, a multifaceted and intricate disease, continues to challenge the scientific community with its complexity and resilience [1,2]. Central to this complexity is the interplay between various molecular actors, among which RHAMM and ncRNAs stand out for their critical roles in tumor biology. RHAMM, a co-receptor for hyaluronan, has been implicated in various cellular processes essential for cancer progression, including cell migration, proliferation, and invasion [3,4]. Its relevance in cancer is underscored by its association with poor prognosis and increased metastatic potential in several cancer types.

Parallelly, ncRNAs, which include miRNAs, lncRNAs, and circRNAs, have emerged as key regulatory molecules in gene expression [5,6]. Their role in cancer encompasses the modulation of oncogenes and tumor suppressors, influencing every stage of cancer development from initiation to metastasis. The ability of ncRNAs to regulate a wide spectrum of biological functions positions them as central figures in post-transcriptional regulation, affecting cellular behavior in the oncogenic context. The interaction between RHAMM and ncRNAs represents a novel and compelling facet of cancer research, offering potential insights into the mechanisms driving cancer progression. This crosstalk could elucidate previously unknown pathways of cancer development, revealing how these interactions contribute to hallmarks of cancer such as sustained proliferative signaling, evasion of growth suppressors, and activation of invasion and metastasis. Understanding these interactions holds promise for identifying new therapeutic targets, enabling the development of more effective, targeted cancer treatments.

Beyond ncRNA-based regulation, recent research has advanced several innovative strategies directly targeting RHAMM to modulate tumor behavior. Function-blocking peptides (e.g., NPI-110, NPI-106, and P15-1) have demonstrated significant anti-fibrotic and potential anti-tumor effects by altering stromal architecture and reducing extracellular matrix remodeling. Additionally, HA oligomers have shown the ability to disrupt RHAMM-mediated signaling complexes, thus impairing cancer cell motility and invasive properties. Anti-RHAMM antibodies have also been developed to block HA binding and inhibit tumor cell migration, further highlighting RHAMM’s therapeutic potential. Importantly, RHAMM-targeted peptide vaccines such as the RHAMM-R3 epitope have emerged as promising immunotherapeutic approaches, capable of inducing strong CD8^+^ T-cell responses, particularly in hematologic malignancies, and enhancing graft-versus-leukemia effects.

This review aims to dissect the current knowledge on the interplay between RHAMM and ncRNAs in cancer, while also exploring therapeutic potentials arising from peptide inhibitors, HA oligomers, antibody-based strategies, and peptide vaccines targeting RHAMM. Collectively, these insights not only deepen the understanding of cancer biology but also pave the way for the development of innovative, targeted therapies, marking a significant advancement in the fight against cancer.

## 2. Structural and Molecular Regulation of RHAMM Expression and Function

### 2.1. Gene Location and Basic Characteristics

RHAMM, alternatively referred to as CD168, is a coiled glycoprotein produced from the HMMR gene [7]. This gene is situated on the 5q33.2-qter region of chromosome 5 in humans and is characterized by its 18 exons along with two initiation codons [8,9].

### 2.2. Transcriptional Regulation via Signaling Pathways

The expression of RHAMM is regulated by a complex network of signaling pathways. HA’s interaction with CD44, the cell surface glycoprotein CD44, activates Protein Kinase C delta (PKCδ), which then phosphorylates the small GTPase Ras-related C3 botulinum toxin substrate 1 (Rac1), triggering the c-Jun N-terminal kinase (JNK) pathway [10]. This leads to the activation of the transcription factors c-Jun and c-Fos, components of the Activator Protein 1 (AP-1) complex, essential for gene transcription. Simultaneously, the mevalonate pathway generates geranylgeranyl pyrophosphate (PP), which is crucial for the activation of Ras homolog family member A (RhoA). RhoA, in turn, activates Yes-associated protein (YAP), a coactivator controlled by phosphorylation and its interaction with the TEA Domain Transcription Factor (TEAD). Transforming growth factor beta-1 (TGF-β1) stimulates SMAD family member 3 (SMAD3), which synergizes with YAP to increase RHAMM transcription [11,12].

The mechanistic target of rapamycin complex 1 (mTORC1) plays a crucial role in the regulation of Serum Response Factor (SRF), a transcription factor that is key for various cellular processes. When mTORC1 is activated, it can lead to the activation of SRF. This activation typically occurs through a cascade of intracellular signaling events, which often include the phosphorylation of downstream targets that directly or indirectly influence SRF’s transcriptional activity [13,14]. This intricate orchestration of PKCδ, mevalonate, RhoA, YAP/TEAD, SMAD3, and mTORC1 pathways finely tunes RHAMM expression, impacting cellular movement and other oncogenic activities.

### 2.3. Post-Transcriptional Events and Cytoplasmic Translation

The transcribed RHAMM pre-mRNA undergoes post-transcriptional processing and moves from the nucleus to the cytoplasm. Given that RHAMM mRNA lacks both a membrane-spanning domain and a signal peptide sequence [15,16], its translation occurs in the cytoplasmic environment.

The human RHAMM protein, after being translated, consists of 725 amino acids with a molecular weight of 84 kDa and undergoes folding into its functional structure [17]. Currently, there is no experimentally determined 3D structure of RHAMM available from X-ray crystallography, NMR, or cryo-electron microscopy. Instead, AlphaFold, a cutting-edge protein structure prediction tool, offers a model estimating the amino acid positions and their confidence scores. This model predicts that the conformation of human RHAMM is predominantly alpha-helical.

### 2.4. Predicted Structure and Functional Domains of RHAMM

The structure of the RHAMM protein encompasses distinct domains integral to its diverse functions. From amino acids 1 to 163, the N-terminal domain features microtubule-binding sites between amino acids 40 to 59 and 76 to 90, highlighting its role in cellular structure and transport. Central to RHAMM is the supercoiled-coil domain, positioned between the N-terminal and C-terminal domains, essential for protein–protein interactions. This facilitates RHAMM’s engagement with cellular receptors such as CD44 and PDGFR, and proteins like BRCA1, TPX2, and ERK1/2, aiding in cell migration and mitotic spindle regulation [18,19,20,21]. Additionally, RHAMM possesses HA binding domains between amino acids 635 to 644 and 657 to 666, emphasizing its function in cell adhesion and signaling [17,22]. The protein concludes with the C-terminal domain around amino acids 715 to 725.

### 2.5. Unconventional Export and Surface Localization

Located primarily within cells, RHAMM is exported to the cell surface via unconventional pathways, which may involve exosomal transport or flippase activity, as well as release from apoptotic cells. Through these unique export mechanisms, RHAMM is able to interact with surface receptors like CD44, enabling its participation in extracellular signaling processes. This pathway underscores the sophisticated cellular strategies for protein localization and function outside the conventional secretion system. The detailed illustration of RHAMM gene expression, protein domains, and exportation is represented in Figure 1.

## 3. The Function of RHAMM in Cancer

### 3.1. Extracellular Functions and ERK1/2 Activation

The HA-binding extracellular functions of RHAMM contribute to normal wound repair in culture and in vivo [23,24], and to the progression of diseases such as arthritis in animal models [25]. Mechanistically, extracellular RHAMM ‘activates’ pro-migration and invasion functions of the transmembrane-adhesion and HA receptor CD44. This RHAMM-regulated activation process results in increased cell-surface expression of CD44 and an increase in the activation of ERK1 and ERK2 (ERK1/2) by CD44 in the presence of HA [18,24]. It was first reported that H-ras transformed cells exhibited characteristics similar to RHAMM overexpressing cells, and that loss-of-function mutation of RHAMM resulted in decreased ERK1/2 activation in Ras mutant cells [26,27]. Subsequent studies demonstrated that RHAMM overexpressing cells exhibit high basal ERK1/2 activation while RHAMM null fibroblasts showed reduced ERK1/2 phosphorylation with no change in total ERK1/2 levels [24,26]. RHAMM has been shown to co-localize with ERK1/2 in breast cancer cells, but the exact mechanism by which RHAMM activates ERK1/2 is currently unknown [18].

### 3.2. Intracellular Roles in Mitosis and Microtubule Regulation

The intracellular activities of RHAMM are essential and varied, playing a pivotal role in cellular mitosis management and maintaining genomic integrity. RHAMM is recognized as a microtubule-associated protein (MAP) that is integral to the structural and functional integrity of mitotic spindles, centrosome dynamics, and the cell division cycle. RHAMM’s interaction with microtubules and centrosomes is critical for ensuring the proper assembly and function of the mitotic spindle, which is essential for accurate cell division. It directly engages with microtubules during both interphase and mitosis, covering their full length, and also associates with centrosomes, collaborating with additional centrosomal proteins during cell division. This ensures the correct assembly of spindles and the successful completion of cell division [28,29]. The disruption of RHAMM, either by suppression or overexpression, can result in significant cell division errors, such as the loss of spindle integrity, formation of multipolar spindles, or induction of a metaphase block, which ultimately can lead to cell death. These outcomes emphasize RHAMM’s essential role in the progression of the cell cycle [29,30,31]. Moreover, RHAMM plays a crucial role in the regulation of microtubule dynamics. It achieves this by interacting with the dynein motor complex and other MAPs, securing the correct positioning and stabilization of spindle poles. Such interactions are vital for the stability of the spindle and the accurate segregation of chromosomes during cell division [31,32]. RHAMM also facilitates the activation of TPX2, crucial for the effective activation of aurora kinase A (AURKA), highlighting its involvement in the formation of microtubules and the assembly of spindles [19]. RHAMM’s interactions extend beyond structural roles, engaging with key proteins involved in cell cycle regulation, such as BRCA1. This interaction plays a role in the amplification of centrosomes and the assembly of spindles, potentially affecting tumor progression and genomic instability. Its role in modulating the G_2_-M transition, through influencing the localization and activation of TPX2 and AURKA, highlights RHAMM’s significant impact on the biology of cancer cells [33,34]. Additionally, RHAMM serves both as an adaptor protein and a transcriptional co-activator. It influences the transcription of genes critical for cell migration and potentially links its intracellular roles to extracellular functions. This includes interactions with hyaluronic acid and modifying the binding affinity between HA and TPX2, which may affect AURKA activity and the cell division cycle [35,36,37,38]. A schematic overview of RHAMM-related signaling cascades and its intracellular roles is illustrated in Figure 2.

## 4. RHAMM Expression in Cancer

### 4.1. Prognostic Significance and Cancer Type-Specific Expression

Contemporary studies have established a correlation between the levels of RHAMM, measured through the total mRNA or protein within tumors, and their prognostic significance. Elevated levels of RHAMM have been associated with a majority of cancer types and often indicate a less favorable prognosis. For instance, higher levels of RHAMM have been noted as a marker for adverse outcomes in various cancers such as breast cancer [39,40,41], multiple myeloma [42], and others, ranging from oral squamous cell carcinoma [43] to gastric cancer [44]. Notably, examinations of tumor histology have shown that RHAMM expression can be diverse within the tumor, with significant expression sometimes confined to specific cancer cell groups [39].

### 4.2. Transcriptomic Insights and Diagnostic Potential

Tools like TNMPlot offer a critical resource for examining the genetic underpinnings of cancer [45]. Using RNA-Seq data from TNMPlot, disparities in RHAMM expression between cancerous tissues and their normal counterparts have been quantified. For example, acute myeloid leukemia (AML) exhibits a pronounced increase in RHAMM expression, with the mean fold change recorded at 13.19 and the median at 22.25. In contrast, pancreatic adenocarcinoma presents a reduction in the mean fold change in RHAMM expression (0.91), with a slightly higher median fold change (1.43), suggesting that RHAMM’s role may vary across different types of cancer. The statistical significance of these variations in RHAMM expression is underscored by Mann–Whitney *p*-values, with some cancers presenting highly significant values, such as colon adenocarcinoma with a *p*-value of 1.3 × 10^−6^. This robust set of data reinforces the potential of RHAMM as a diagnostic biomarker and as a target for cancer therapy. The summarized expression of RHAMM across various cancers is detailed in Table 1.

### 4.3. Context-Dependent Roles and Therapeutic Relevance

The impact of RHAMM on cancer progression is possibly influenced by the specific subtype of cancer cells expressing RHAMM and their location within the tumor, such as regions associated with invasion or metastasis. Interestingly, there are instances where RHAMM functions as a tumor suppressor, such as its genomic reduction being linked to the development and progression of malignant peripheral nerve sheath tumors [46]. Given RHAMM’s capacity for versatile interactions leading to cell-specific outcomes, the contrasting roles of RHAMM across different studies underscore the necessity for comprehensive research to inform targeted therapy strategies. Despite the complexity of RHAMM’s functions, its limited expression in normal physiological conditions, correlation with poor cancer outcomes, and involvement in processes like inflammation and fibrosis [47,48], which can aggravate cancer progression, make it a compelling candidate for therapeutic targeting. Moreover, pioneering work with RHAMM peptide mimetics that disrupt HA-RHAMM signaling on the cell surface has revealed that this extracellular interaction may orchestrate the multifaceted functions of RHAMM by initiating and activating a cascade of downstream effects.

## 5. Therapeutic Strategies Targeting RHAMM in Cancer

### 5.1. Regulation of RHAMM by Non-Coding RNAs

Non-coding RNAs (ncRNAs) such as miRNAs, lncRNAs, and circRNAs are key regulators of cancer biology through modulation of gene expression [49,50,51,52,53]. In particular, miRNAs mediate gene silencing via the RISC complex, while lncRNAs and circRNAs influence these processes by chromatin remodeling or acting as miRNA sponges [54,55,56,57,58].

LncRNAs regulate gene expression at multiple levels, including chromatin remodeling and transcription, and act as oncogenes or tumor suppressors, influencing proliferation, apoptosis, metastasis, and drug resistance [59,60,61]. They are transcribed by RNA polymerases I–III and undergo post-transcriptional modifications, but are not translated into proteins [62,63].

CircRNAs are generated via back-splicing mediated by RNA polymerase II, forming stable loop structures without 5′ to 3′ polarity or polyadenylated tails [64,65,66]. They can function as molecular scaffolds or miRNA sponges, affecting cancer cell proliferation, metastasis, and apoptosis [67].

In cancer, the RHAMM and ncRNA network plays a crucial role by influencing tumor progression, metastasis, and response to treatment. This network involves complex interactions between ncRNAs with RHAMM. Understanding this network provides insights into cancer biology, offering potential targets for novel therapeutic strategies and biomarkers for early diagnosis and prognosis assessment.

First, let us explore the types of miRNAs that regulate the stability of RHAMM mRNA and examine recent studies on lncRNAs and circRNAs involved in sponging these miRNAs. This highlights the post-transcriptional regulation of RHAMM by ncRNAs and their therapeutic potential in cancer. Qihao Wang et al.’s study demonstrated that microRNA let-7c-5p suppresses the growth and spread of lung adenocarcinoma (LUAD) by inversely targeting and downregulating RHAMM mRNA, suggesting the let-7c-5p/HMMR interaction as a promising therapeutic target for LUAD [68].

Wei Li and colleagues discovered that LUAD (lung adenocarcinoma) exhibits elevated RHAMM levels linked to negative patient outcomes [69]. The study reveals that miR-34a-5p, a tumor-suppressing miRNA with reduced expression in LUAD, targets RHAMM. The lncRNA HCG18 captures miR-34a-5p, thereby upregulating RHAMM and advancing LUAD, underscoring the pivotal role of the HCG18/miR-34a-5p/RHAMM pathway in promoting LUAD tumor growth.

Yong Peng and colleagues demonstrate that hsa_circ_0005273 in breast cancer (BC) functions as an miR-509-3p sponge to upregulate RHAMM expression, promoting BC’s malignant characteristics [70]. Knockdown of circ_0005273 reduced BC cell proliferation, migration, invasion, and enhanced apoptosis, indicating the therapeutic potential of targeting the circ_0005273/miR-509-3p/HMMR axis in BC.

Fang He and colleagues revealed that hsa-miR-411-5p plays a significant role in ovarian cancer by targeting RHAMM to inhibit cell proliferation, migration, and invasion [71]. This discovery positions miR-411-5p as both a potential biomarker and therapeutic target, emphasizing its crucial impact on ERK1/2 activity and the behavior of ovarian cancer cells. The findings indicate that miRNAs, specifically let-7c-5p, miR-34a-5p, miR-509-3p, and hsa-miR-411-5p, play a key role in the degradation of RHAMM mRNA. Notably, miR-34a-5p and miR-509-3p are regulated by lncRNA HCG18 and circRNA hsa_circ_0005273, respectively. Both HCG18 and hsa_circ_0005273 act to increase RHAMM expression, thereby contributing to tumor growth and increased cell proliferation.

HMMR-AS1, an lncRNA, is transcribed from the opposite strand of the HMMR gene, which encodes the RHAMM receptor. Exploring the intricate dynamics between HMMR-AS1 and its regulatory impact on cancer progression underscores the nuanced interplay of genetic elements in tumor biology. These investigations illuminate the multifaceted roles of HMMR-AS1 in enhancing oncogene activity, revealing a complex network of interactions that are crucial for tumor growth and metastasis. These findings highlight HMMR-AS1’s oncogenic role and its potential as a target in RHAMM-driven cancers.

This HMMR-AS1 exhibits a positive correlation with RHAMM expression, suggesting a synergistic interaction that enhances their expression levels and indicates their collaborative impact on cellular functions. This interaction potentially contributes to disease progression, particularly in types of cancer where high RHAMM expression plays a crucial role [72]. Junyang Li and colleagues have demonstrated the potential of targeting lncRNA HMMR-AS1 in glioblastoma (GBM) [73]. Their research shows that overexpression of HMMR-AS1 in GBM stabilizes RHAMM mRNA, thereby enhancing tumor cell proliferation, migration, and invasion. Conversely, inhibiting HMMR-AS1 not only diminishes these processes but also increases GBM cell radiosensitivity by altering DNA repair proteins, positioning HMMR-AS1 as a promising target for GBM therapy. Yong Cai and his team have illustrated how HMMR-AS1 accelerates the progression of lung adenocarcinoma (LUAD) by manipulating the miR-138/SIRT6 signaling pathway [74]. This highlights its significant contribution to the proliferation and spread of cancer cells. Furthermore, Xu Wang and his team discovered that the exosomal lncRNA HMMR-AS1 modulates hepatocellular carcinoma (HCC) progression by influencing macrophage polarization via the miR-147a/ARID3A axis in a hypoxic environment [75]. This suggests its viability as a therapeutic target for HCC. In addition, Xi’an Bao and colleagues have uncovered that sevoflurane (Sev) inhibits glioma progression through the regulatory network of HMMR-AS1/miR-7/CDK4 [76]. This study illuminates Sev’s potential mechanism of action on glioma cells by influencing cell growth, invasion, and the formation of colonies through the interaction of lncRNA HMMR-AS1 with miR-7 and CDK4. It suggests a novel therapeutic approach to managing glioma. Collectively, these findings underscore the critical role of HMMR-AS1 in modulating cancer progression through various pathways and interactions. Its association with cancer cell proliferation, migration, and invasion, along with the impact on cellular radiosensitivity and signaling pathways, highlights its potential as a target for innovative cancer therapies. This research contributes significantly to the evolving landscape of cancer biology and therapeutics, offering new insights into the complex mechanisms driving cancer progression and providing hope for more effective treatments in the future.

Subsequently, the focus will be on ncRNAs induced by high RHAMM expression and their influence on the behaviors of cancer cells, shedding light on their contribution to cancer progression and their potential as targets for therapy. Huaxin Zhu and colleagues highlighted RHAMM’s upregulation in gliomas, forming a crucial endogenous RNA (ceRNA) network involving HEELPAR lncRNA, hsa-let-7i-5p miRNA, and Ribonucleotide reductase2 (RRM2), significantly affecting glioma biology [77]. RRM2, particularly, enhances glioblastoma cell proliferation, migration, and invasion, with BRCA1-regulated expression aiding in stress resistance and tumorigenicity [68,69]. Enrichment analyses suggest RRM2’s role in glioma via cell cycle and proliferation pathways, indicating RHAMM and RRM2’s joint contribution to glioma progression through cell cycle regulation. This analysis provides valuable insights into the role of ncRNAs in cancer progression, particularly through the ceRNA network involving RHAMM, HEELPAR lncRNA, hsa-let-7i-5p miRNA, and RRM2. While the study by Huaxin Zhu and colleagues offers significant findings on glioma biology and suggests potential therapeutic targets, it also underscores the need for further research. The limitation of having a single study emphasizes the necessity for additional investigations to validate these results and explore the broader implications of RHAMM and its associated ncRNA network in cancer. The regulatory mechanisms involving HMMR-AS1 lncRNA and ncRNAs in RHAMM mRNA regulation and cancer progression are summarized in Figure 3, with a detailed table provided in Table 2.

The use of ncRNAs to regulate RHAMM offers a promising strategy for cancer therapy. MiRNAs, lncRNAs, and circRNAs are key regulators of gene expression and significantly influence RHAMM-driven processes such as cell migration, proliferation, and invasion. Tumor-suppressive miRNAs can be engineered to target RHAMM mRNA, reducing its expression and tumor aggressiveness. In contrast, lncRNAs and circRNAs often act as sponges for these miRNAs, preventing them from inhibiting RHAMM. Blocking these sponges can restore miRNA activity and suppress RHAMM expression, which is particularly valuable in cancers with high RHAMM levels linked to poor prognosis.

MiRNAs naturally target RHAMM mRNA via the RISC complex, leading to degradation or translational repression. However, their clinical application is limited by instability and off-target effects. Chemical modifications, such as 2′-O-methyl (2′-OMe) groups [78,79] and locked nucleic acids (LNAs) [80,81], enhance miRNA stability and specificity. These modifications improve binding affinity, extend half-life, and reduce unintended gene silencing, making miRNA-based therapies more effective and safer.

Efficient delivery systems are also essential. Nanoparticles, liposomes, and virus-like particles can protect miRNAs, enhance cellular uptake, and enable targeted release in tumor tissues. Functionalizing these carriers with ligands or antibodies improves specificity, leading to higher drug accumulation at tumor sites and fewer systemic side effects [82,83,84,85]. Additionally, non-particulate systems using aptamers or modified nucleic acids offer alternative targeted delivery options [86,87,88].

In summary, combining engineered miRNAs with chemical modifications and advanced delivery systems embodies precision medicine, offering a targeted, effective approach with minimal off-target effects. Further research and clinical validation are needed to fully realize this strategy’s potential.

### 5.2. Function-Blocking RHAMM Peptides

RHAMM is a multifunctional protein that contributes significantly to cell motility, invasion, and proliferation in various cancers, including breast, liver, colorectal, and lung cancers [18,89]. Overexpression of RHAMM has been consistently associated with poor prognosis and enhanced metastatic potential [90]. While RHAMM is well recognized for its roles in cancer cell behavior, it also plays an integral role in fibrosis through its regulation of fibroblast activation and extracellular matrix (ECM) remodeling [28].

Recent studies using function-blocking RHAMM peptides, such as NPI-110 and NPI-106, demonstrated potent anti-fibrotic effects in bleomycin-induced murine models of systemic sclerosis. Specifically, NPI-110 treatment significantly reduced dermal thickness, collagen accumulation, and expression of profibrotic markers including TGF-β1, c-Myc, Col1a1, and Col3a1. Furthermore, NPI-110 enhanced the expression of antifibrotic adipokines such as adiponectin and perilipin, suggesting its capacity to modulate stromal environments [91].

In addition to these peptides, an HA-mimetic peptide named P15-1 has been developed to competitively block HA–RHAMM interactions. P15-1 is a 15-amino-acid peptide identified via phage display that mimics RHAMM’s HA-binding domain and specifically binds HA with a dissociation constant around 10^−7^ M. In vitro, P15-1 effectively suppressed fibroblast migration, blocked HA-oligosaccharide-induced signaling, and reduced myofibroblast differentiation and TGF-β1 expression. Crucially, in vivo data from full-thickness excisional wound models in rats showed that topical P15-1 application significantly reduced macrophage and fibroblast infiltration, blood vessel density, collagen I deposition, TGF-β1, and α-SMA levels, while increasing tenascin-C, thus promoting a regenerative, scar-reduced phenotype. These results demonstrate that P15-1 not only interferes with HA–RHAMM signaling but also reprograms the ECM and inflammatory milieu in animal models [92].

Fibrosis and cancer share several mechanistic pathways, particularly the involvement of TGF-β signaling, activation of myofibroblasts, and excessive ECM deposition. In cancer, cancer-associated fibroblasts (CAFs) facilitate tumor progression by creating a dense fibrotic stroma, promoting epithelial–mesenchymal transition (EMT), and mediating immune evasion [93,94]. Targeting RHAMM in this context could potentially disrupt CAF activation and ECM remodeling, thereby attenuating tumor growth and metastatic dissemination.

Moreover, RHAMM inhibition may enhance tumor immune surveillance. Dense stromal barriers hinder immune cell infiltration, limiting the efficacy of immunotherapies such as immune checkpoint inhibitors [95]. By reducing stromal fibrosis and normalizing the tumor microenvironment, RHAMM-targeting peptides could synergize with immunotherapies to improve anti-tumor immune responses.

Although NPI-110 and NPI-106 have not yet been evaluated in cancer models, pre-clinical studies on RHAMM silencing or HA–RHAMM interaction blockade have shown promising results in suppressing tumor invasion and metastasis [24,96]. Therefore, the anti-fibrotic RHAMM-targeting peptides represent a promising therapeutic strategy to modulate tumor stroma and potentially improve cancer treatment outcomes. Further investigations using in vivo cancer models are warranted to validate these hypotheses and to assess their combinatorial potential with existing therapies.

### 5.3. HA Oligomers as RHAMM Inhibitors

Low-molecular-weight HA oligomers have been investigated as potential modulators of RHAMM-mediated signaling in cancer and fibrosis. Unlike large, high-molecular-weight HA, these oligomers can interfere with HA–RHAMM and HA–CD44 interactions, leading to context-dependent effects on cell behavior. In fibrosarcoma HT1080 cells, HA oligomers were shown to reduce RHAMM-dependent adhesion to fibronectin and inhibit cell migration by suppressing ERK1/2 and FAK phosphorylation, suggesting an inhibitory role in tumor cell motility [97]. Similarly, studies have demonstrated that HA oligomers disrupt HA-mediated signaling complexes involving RHAMM and CD44, thereby attenuating pro-survival and pro-migratory pathways in cancer cells [98].

However, it is important to note that HA oligomers can also promote certain signaling events depending on concentration and cellular context, sometimes inducing transient ERK1/2 activation and promoting wound repair responses [24,98]. Despite this duality, the ability of HA oligomers to competitively inhibit HA binding to RHAMM highlights their potential as therapeutic agents to modulate tumor stroma and reduce cancer cell invasiveness. Further in vivo studies are necessary to validate their specific anti-tumor effects and to clarify their functional outcomes across different cancer types.

### 5.4. RHAMM-Targeting Antibodies

Early studies using polyclonal antibodies against RHAMM demonstrated inhibition of HA binding and reduced cell motility, highlighting the potential of antibody-based blockade of RHAMM-mediated signaling [99]. Further evidence supporting antibody-based RHAMM targeting was provided by Pilarski et al., who demonstrated that a monoclonal antibody against RHAMM markedly suppressed HA-induced locomotion of malignant B cells derived from patients with hairy cell leukemia [100]. This study highlighted the ability of anti-RHAMM antibodies to interfere with RHAMM-dependent cytoskeletal reorganization and migration, suggesting potential utility in limiting tumor invasiveness.

Despite these encouraging pre-clinical results, further in vivo studies and clinical evaluations are necessary to establish the therapeutic efficacy, optimal delivery strategies, and safety profiles of RHAMM-targeting antibodies across different cancer types. An overview of RHAMM-targeting peptides, HA oligomers, and antibodies, including their experimental models and observed biological effects, is illustrated in Figure 4 and summarized in Table 3.

### 5.5. RHAMM-Derived Peptide Vaccines

The quest for more effective cancer treatments has led to significant interest in immunotherapy, particularly strategies that target specific tumor-associated antigens [101,102]. This review examines the landscape of RHAMM-targeted immunotherapies, from peptide vaccines to dendritic cell (DC)-based treatments, evaluating their efficacy and the challenges encountered in leveraging RHAMM for cancer immunotherapy.

Research into RHAMM as a target for immunotherapy has revealed a promising avenue for treating various forms of leukemia, with studies suggesting the potential for both polyvalent vaccinations and DC-based therapies. The identification of several leukemia-associated antigens (LAAs), including RHAMM, which induce humoral immune responses in acute and chronic myeloid leukemia, underscores the feasibility of such immunotherapeutic strategies [103]. Further investigations have shown that DCs derived from acute myeloid leukemia (AML) patients inherently express RHAMM alongside other LAAs, highlighting their potential in vaccine-based treatments [104].

The discovery of RHAMM-specific CD8+ T-cell epitopes has marked a significant milestone, indicating RHAMM’s viability as an immunotherapy target and leading to the initiation of a clinical trial with the R3 peptide [105]. In early-stage B-cell chronic lymphocytic leukemia (B-CLL), vaccination with allogeneic DCs pulsed with tumor lysates or apoptotic bodies has been demonstrated as both feasible and safe, eliciting specific cytotoxic T lymphocyte responses against RHAMM [106]. Similarly, chronic myeloid leukemia (CML) cells have been shown to express RHAMM/CD168, inducing specific CD8+ T-cell responses and suggesting the potential for RHAMM-R3-directed effector T cells in immunotherapy [107].

Notably, the RHAMM-derived peptide R3 has been shown to induce specific CD8+ T cell responses in both healthy donors and CML patients following allogeneic stem cell transplantation, hinting at the peptide’s capability to enhance graft-versus-leukemia effects [108]. In a mouse glioma model, vaccination with DCs transfected with modified RHAMM mRNA significantly prolonged survival, demonstrating potent immunological antitumor effects [109]. Similarly, vaccination with autologous DCs pulsed with B-CLL cell lysates in early-stage B-CLL patients has been shown to increase specific CD8+ T cells against leukemia antigens like RHAMM, indicating a shift towards an anti-leukemia response [110].

However, high-dose RHAMM-R3 peptide vaccination, while inducing immunological responses and positive clinical effects in patients with acute myeloid leukemia, myelodysplastic syndrome, and multiple myeloma, did not necessarily enhance these responses with higher doses, suggesting a complex relationship between dosage and efficacy [111]. Phase I clinical trials have further corroborated the safety and efficacy of R3 peptide vaccination in eliciting specific CD8+ T-cell responses against RHAMM in HLA-A2+ CLL patients [112]. A subsequent phase I/II clinical trial revealed that RHAMM-derived peptide vaccination not only induced R3-specific cytotoxic T cells but also resulted in profound changes in various T-cell subsets and cytokines, showcasing the potential for T-cell-based immunotherapy in CLL [113].

Contrastingly, RHAMM has been critiqued for not meeting the ideal criteria as a target antigen in immunotherapy for AML, due to its expression in leukemic stem cells being comparable to that in healthy hematopoietic stem cells and its significant upregulation in proliferating cells and activated T cells [114]. This challenge notwithstanding, the potential incorporation of the tumor-associated antigen RHAMM into monocyte-derived dendritic cells (mo-DCs) through mRNA electroporation for cancer immunotherapy has been explored, revealing that classical mo-DCs already naturally express and present RHAMM at levels sufficient to activate RHAMM-specific T cells in AML patients, suggesting that existing immunotherapy approaches may already inadvertently target RHAMM [115]. The information regarding RHAMM-targeted immunotherapies is consolidated in Table 3.

DC-based peptide delivery emerges as a potent approach in cancer immunotherapy, leveraging autologous DCs loaded with tumor antigens to enhance tumor-specific T-cell responses, demonstrating observable antitumor immunity in various studies [116]. By incorporating the RHAMM R3 peptide into the immunization strategy, DCs induce activation of cytotoxic T cells (Tc cells) and prime them to recognize and destroy cancer cells expressing the RHAMM antigen. The mature DCs loading the RHAMM R3 peptide through MHC class I molecules present to Tc cells, a critical step for the specific targeting of cancer cells. The interaction between the T-cell receptor (TCR) on Tc cells and the peptide–MHC complex on DCs triggers the activation and proliferation of Tc cells, gearing them towards recognizing and attacking cancer cells exhibiting the RHAMM peptide. The activated Tc cells migrate to tumor sites where they engage with cancer cells presenting the RHAMM R3 antigen, leading to the release of cytotoxic granules that induce cancer cell apoptosis. This targeted killing spares healthy cells that do not present the RHAMM antigen, minimizing collateral damage and improving the therapy’s safety profile. The mechanism of the RHAMM R3 peptide immunovaccine is depicted in Figure 5.

The investigation into RHAMM-targeted immunotherapies has showcased both their promise and challenges, particularly in leukemia where they have successfully elicited immune responses against leukemic cells. The discovery of RHAMM-specific T cells in AML patients and the imperative for precision in targeting RHAMM to prevent harm to healthy cells highlight the opportunity to broaden the application of these therapies to other cancers. Such expansion necessitates in-depth research to fully grasp RHAMM’s function in various malignancies and to craft vaccines capable of accurately targeting a wider range of cancer cells. The potential integration of RHAMM-targeted approaches with combination therapies or advanced targeting technologies underscores the need for ongoing innovation in research to leverage RHAMM’s full therapeutic promise in oncology. The specificity of these immunotherapies, while presenting a unique opportunity, also poses significant challenges, emphasizing the critical need for further exploration beyond leukemia. This calls for additional studies aimed at understanding RHAMM’s role in different cancers and developing precise vaccines, ensuring the evolution of RHAMM-targeted immunotherapies and their significant future impact across oncology. Among RHAMM-targeted therapeutic approaches, peptide inhibitors such as P15 or NPI-110 directly block HA-RHAMM interactions and offer specificity with relatively low immunogenicity, though they suffer from rapid degradation and limited in vivo stability. Monoclonal antibodies provide strong target affinity and can recruit immune effector functions, yet may carry a risk of off-target effects and high production costs. RHAMM-based vaccines aim to elicit long-term immune responses, with early clinical trials showing potential, especially in hematologic malignancies, though their efficacy can vary depending on the patient’s immune background and HLA type. These strategies each have distinct advantages and limitations that should be considered in therapeutic design.

The concept and mechanisms of vaccines derived from RHAMM peptides are illustrated in Figure 5 and systematically summarized in Table 4.

## 6. Conclusions

This review emphasizes the pivotal role of RHAMM in cancer progression and its complex regulation by ncRNAs. Beyond genetic and epigenetic control, various novel RHAMM-targeting strategies, including function-blocking peptides, HA oligomers, and anti-RHAMM antibodies, offer promising opportunities to disrupt tumor stroma, reduce invasiveness, and potentially enhance immune responses. Notably, RHAMM-targeted peptide vaccines have shown the ability to elicit potent, specific CD8^+^ T-cell responses and demonstrate efficacy in hematologic cancers, suggesting their applicability in broader cancer immunotherapy. Despite its promise as a cancer therapeutic target, RHAMM has notable limitations. First, its multifunctional nature and involvement in both extracellular and intracellular signaling complicate the prediction of therapeutic outcomes. Second, RHAMM expression can vary significantly among tumor types and even within different regions of a single tumor, potentially reducing the efficacy of RHAMM-targeted therapies. Moreover, the lack of a well-defined 3D crystal structure and its unconventional secretion mechanisms hinder drug development and delivery optimization. These challenges highlight the need for further research to refine RHAMM-based interventions. Overall, targeting RHAMM, together with understanding its interaction with ncRNAs and immunologic mechanisms, may pave the way for more precise and effective cancer therapies that improve patient outcomes and survival.

## Figures and Tables

**Figure 1 ijms-26-07198-f001:**
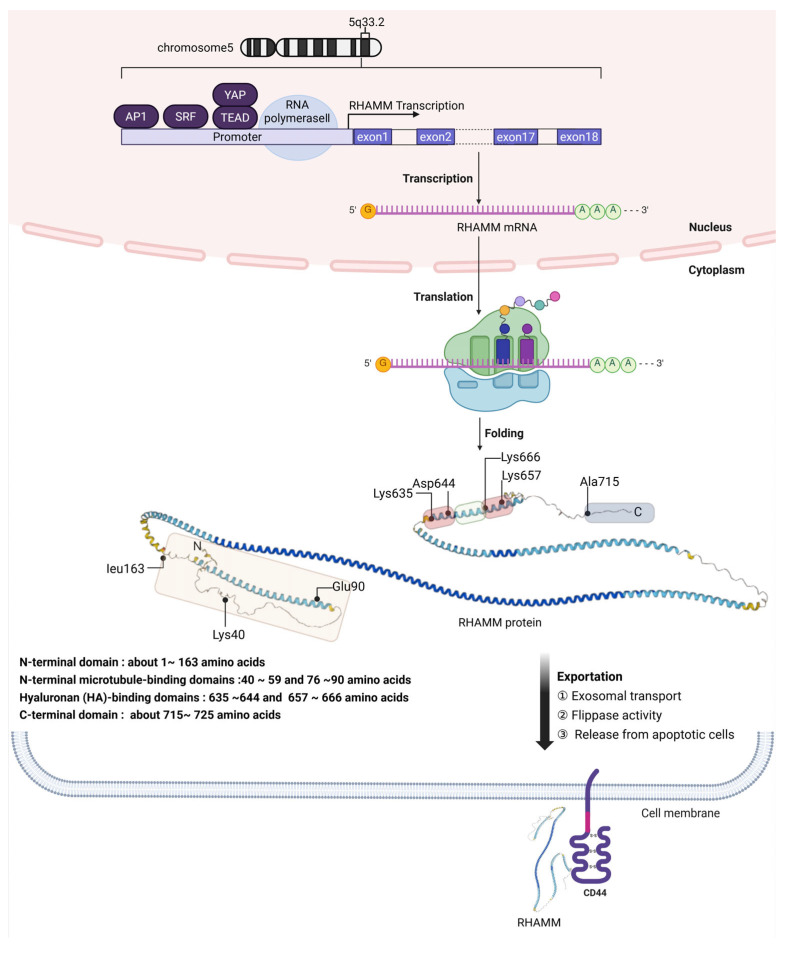
Schematic overview of RHAMM gene expression and protein localization. This detailed diagram presents the multifaceted process of RHAMM (CD168) gene expression, beginning with its chromosomal location on human chromosome 5 at the 5q33.2 region, through the complex transcriptional regulation involving transcription factors like AP1, SRF, YAP, and TEAD. It depicts the journey of RHAMM pre-mRNA from its synthesis in the nucleus, where it lacks traditional secretory signal peptides, to its translation in the cytoplasm into a predominantly alpha-helical structure. The figure outlines the protein’s functional domains: N-terminal microtubule-binding domains at amino acids 40–59 and 76–90, HA-binding domains at amino acids 635–644 and 657–666, and the C-terminal domain near amino acids 715–725. The unconventional export of RHAMM to the cell surface, potentially via mechanisms such as exosomes or flippases, culminates in its interaction with cell surface receptors like CD44, emphasizing the protein’s role in cell signaling and migration. The secondary structure of the RHAMM protein was downloaded from the AlphaFold database at https://alphafold.ebi.ac.uk/entry/O75330 (accessed on 30 May 2025).

**Figure 2 ijms-26-07198-f002:**
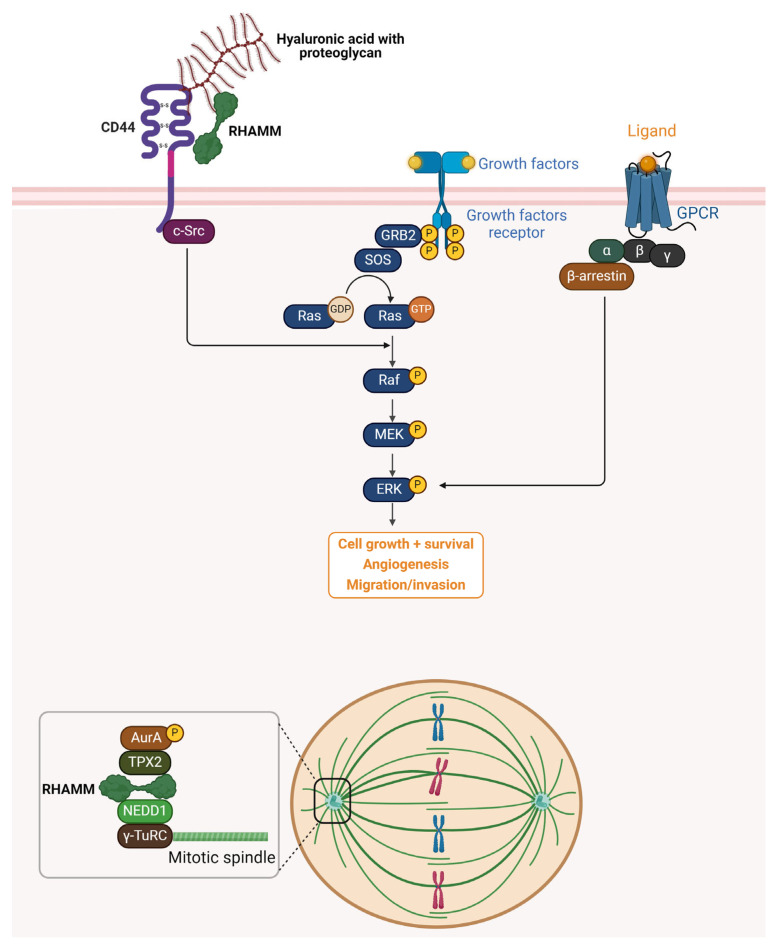
Schematic representation of RHAMM signaling pathways and intracellular functions. Extracellularly, RHAMM interacts with HA and proteoglycans through its association with the cell surface receptor CD44, facilitating the activation of downstream signaling cascades such as the Ras–Raf–MEK–ERK pathway. This activation promotes cell growth, survival, and angiogenesis, and enhances migration and invasion capabilities. Additionally, growth factor receptors and G-protein-coupled receptors (GPCRs) further modulate ERK signaling through GRB2, SOS, and β-arrestin-mediated pathways. Intracellularly, RHAMM functions as a microtubule-associated protein (MAP), playing a pivotal role in mitotic spindle assembly and stabilization. It forms complexes with TPX2, Aurora kinase A (AurA), NEDD1, and γ-TuRC, ensuring proper spindle formation and chromosome segregation during cell division. RHAMM also interacts with proteins such as BRCA1, influencing centrosome dynamics, genomic stability, and cell cycle progression. Together, these extracellular and intracellular roles underscore RHAMM’s multifaceted contributions to tumor progression and metastasis.

**Figure 3 ijms-26-07198-f003:**
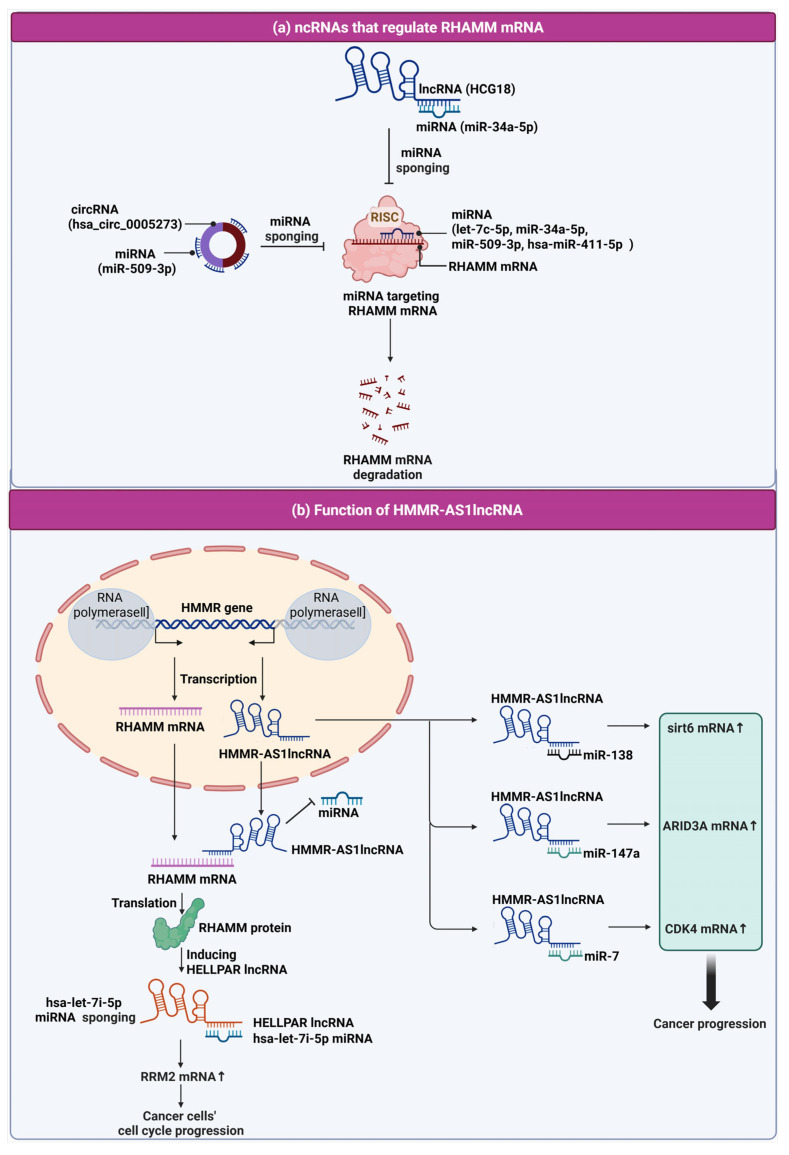
Schematic representation of the regulatory mechanisms involving HMMR-AS1 lncRNA and ncRNAs in RHAMM mRNA regulation and cancer progression. (**a**) NcRNAs that regulate RHAMM mRNA. Included in this regulatory network are lncRNA HCG18 and circRNA (hsa_circ_0005273), both of which function as miRNA sponges. LncRNA HCG18 binds to miR-34a-5p, and hsa_circ_0005273 interacts with miR-509-3p, effectively preventing these miRNAs from targeting RHAMM mRNA for degradation. Additionally, miRNAs such as let-7c-5p, miR-34a-5p, miR-509-3p, and hsa-miR-411-5p are depicted as part of the RISC complex, which targets RHAMM mRNA, leading to its degradation. (**b**) Function of HMMR-AS1 lncRNA in cancer. This panel depicts the transcription of the HMMR gene and the lncRNA HMMR-AS1. HMMR-AS1 is transcribed in the opposite direction to RHAMM mRNA from the HMMR gene. HMMR-AS1 lncRNA plays a multifaceted role in regulating cancer progression by sponging various miRNAs. It binds to miR-138, thereby influencing sirt6 mRNA levels; it interacts with miR-147a, affecting ARID3A mRNA levels; and it associates with miR-7, which in turn regulates CDK4 mRNA levels. These interactions promote cancer cell cycle progression by upregulating mRNA transcripts that are pivotal for cancer cell proliferation and survival. Moreover, the figure shows HMMR-AS1 lncRNA’s impact on RHAMM mRNA stability and translation into RHAMM protein, which is involved in cancer cell cycle progression. The diagram also highlights the action of hsa-let-7i-5p miRNA sponging by HELLPAR lncRNA, which consequently affects RRM2 mRNA and furthers cancer cell cycle progression, demonstrating the intricate web of ncRNA interactions that contribute to cancer development and potential targets for therapeutic intervention.

**Figure 4 ijms-26-07198-f004:**
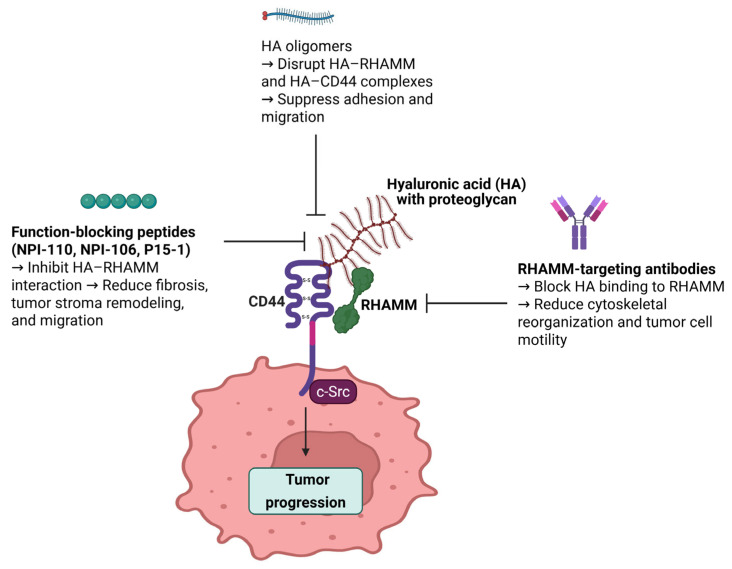
Schematic representation of therapeutic strategies targeting RHAMM to inhibit tumor progression. This figure illustrates three main approaches to inhibit RHAMM-mediated tumor progression. Function-blocking peptides (NPI-110, NPI-106, and P15-1) inhibit HA–RHAMM interactions, thereby reducing fibrosis, tumor stroma remodeling, and cell migration. HA oligomers disrupt HA–RHAMM and HA–CD44 complexes, leading to suppressed adhesion and migration of cancer cells. RHAMM-targeting antibodies block HA binding to RHAMM, ultimately reducing cytoskeletal reorganization and tumor cell motility. Together, these strategies interfere with the HA–RHAMM signaling axis, contributing to decreased tumor progression and potentially enhancing the effectiveness of combinatorial cancer therapies.

**Figure 5 ijms-26-07198-f005:**
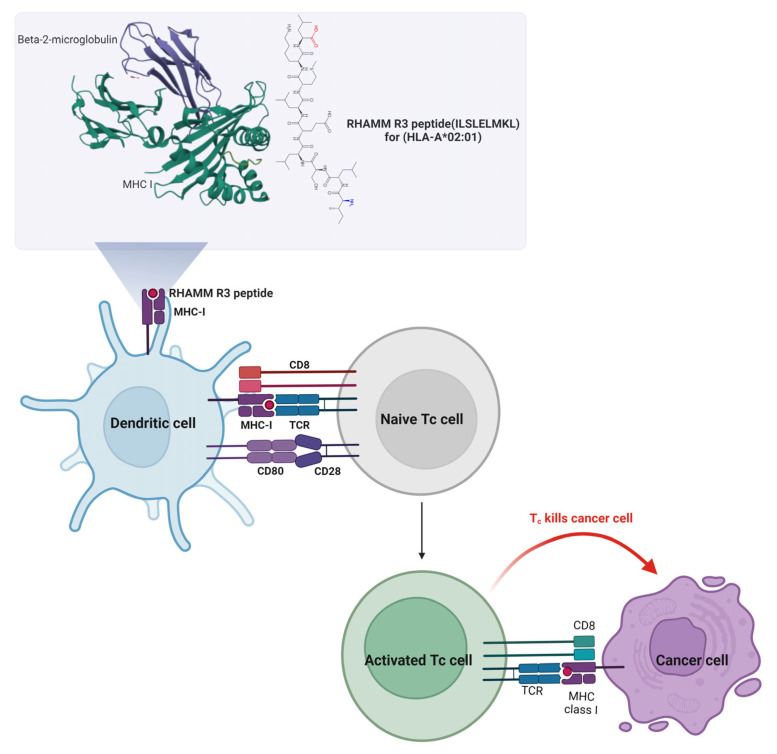
Mechanism of RHAMM R3 peptide-based DC vaccine activating cytotoxic T cells to target cancer cells. This figure illustrates the mechanism by which the RHAMM R3 peptide (ILSLELMKL) restricted to HLA-A*02:01 is presented by DCs to activate cytotoxic T cells (Tc cells). In the upper panel, the RHAMM R3 peptide is shown bound to MHC class I molecules, supported by beta-2-microglobulin. DCs load and present this peptide via MHC I, engaging with naive Tc cells through the T-cell receptor (TCR) and co-stimulatory molecules (CD80–CD28 and CD8). This interaction triggers activation and proliferation of Tc cells. Activated Tc cells migrate to tumor sites and recognize cancer cells expressing RHAMM R3 antigen via MHC I, leading to targeted killing through the release of cytotoxic granules. This approach aims to induce tumor-specific immune responses while sparing normal cells lacking RHAMM expression, thus enhancing therapeutic specificity and minimizing off-target effects.

**Table 1 ijms-26-07198-t001:** The analysis of RHAMM expression in both healthy and cancerous tissues utilized RNA sequencing and chip-based data, sourced from TNMPlot. The platform chosen for this examination comprised tumor samples alongside their corresponding normal tissues for comparison.

RNA-Seq Data			
Tissue	Mann–Whitney *p* ^a^	Fold Change Mean ^b^	Fold Change Median ^c^
**Colon Adenocarcinoma**	1.3 × 10^−6^	3.00	2.94
**Bladder Urothelial Carcinoma**	1.68 × 10^−4^	5.51	15.00
**Breast Invasive Carcinoma**	8.23 × 10^−20^	8.03	9.29
**AML**	1.98 × 10^−65^	13.19	22.25
**Cervical Squamous Cell Carcinoma and Endocervical Adenocarcinoma**	1.81 × 10^−1^	33.29	91.80
**Cholangiocarcinoma**	9.09 × 10^−3^	33.64	13.18
**Esophageal Carcinoma**	1.43 × 10^−2^	4.09	3.30
**Head and Neck Squamous Cell Carcinoma**	7.86 × 10^−7^	2.84	2.75
**Kidney Chromophobe**	2.85 × 10^−2^	3.80	1.52
**Kidney Renal Clear Cell Carcinoma**	8.43 × 10^−10^	2.78	2.72
**Kidney Renal Papillary Cell Carcinoma**	1.08 × 10^−5^	6.82	4.79
**Liver Hepatocellular Carcinoma**	2.29 × 10^−9^	13.23	15.80
**Lung Adenocarcinoma**	1.05 × 10^−10^	8.50	9.62
**Lung Squamous Cell Carcinoma**	1.22 × 10^−9^	12.41	11.79
**Pancreatic Adenocarcinoma**	8.55 × 10^−1^	0.91	1.43
**Pheochromocytoma and Paraganglioma**	4.23 × 10^−1^	2.60	3.17
**Prostate Adenocarcinoma**	1.52 × 10^−7^	3.11	3.59
**Rectum Adenocarcinoma**	3.3 × 10^−2^	2.24	3.38
**Gene Chip Data**			
**Breast**	2 × 10^−10^	4.02	2.57
**CNS**	7.89 × 10^−1^	6.72	0.86
**Colon**	2.51 × 10^−19^	2.18	2.07
**Gastric**	1.97 × 10^−28^	2.25	2.17
**Kidney**	6.7 × 10^−14^	2.05	1.54
**Liver**	5.01 × 10^−21^	6.27	7.77
**Lung**	1.45 × 10^−40^	5.13	6.56
**Lymphoid**	1.07 × 10^−1^	0.23	0.20
**Neural**	4.23 × 10^−1^	4.08	2.47
**Esophageal**	2.92 × 10^−8^	6.23	5.02
**Oral cavity**	1 × 10^−1^	3.56	5.01
**Ovarian**	5.91 × 10^−2^	3.70	3.36
**Pancreas**	2.39 × 10^−6^	2.29	2.12
**Parathyroid**	1 × 10^0^	0.90	0.90
**Prostate**	1.51 × 10^−4^	2.96	2.94
**Skin**	9.23 × 10^−3^	1.79	1.99
**Soft Tissue**	1 × 10^0^	1.06	1.15
**Thyroid**	2.13 × 10^−3^	1.13	1.82
**Uterus**	1.32 × 10^−1^	1.90	2.33

^a^: Denotes the *p*-value derived from the Mann–Whitney U test, a nonparametric method employed to assess the significance of variance between two distinct groups. This statistical test is particularly useful for comparing differences in gene expression levels, such as RHAMM, between normal and cancerous tissues. ^b^: Reflects the mean fold change in expression of the RHAMM gene when contrasting cancerous tissues with normal tissues. The fold change is utilized to describe the extent of variation in a certain measure across two different states; for gene expression analysis, it is often applied to express the difference in a gene’s expression level from a baseline (normal tissue) to an experimental state (cancerous tissue). ^c^: Analogous to the mean fold change, this value corresponds to the median fold change in RHAMM gene expression observed between the two states under comparison. The median represents the central point of a data set arranged in ascending order and serves as an indicator of central distribution that is more robust to anomalous values and distribution skewness than the mean.

**Table 2 ijms-26-07198-t002:** Regulation of RHAMM by ncRNAs in various cancers.

ncRNA Type	Cancer Type	Cell Line/Animal Model	Remarks	Ref.
**let-7c-5p miRNA**	Lung Adenocarcinoma (LUAD)	LUAD cells, HLF-a cells	let-7c-5p miRNA negatively regulates RHAMM, reducing cell proliferation, migration, and invasion.	[68]
**lncRNA (HCG18), miRNA** **(miR-34a-5p)**	LUAD	LUAD cell lines (A549, H1299, Calu3, and HCC827)	HCG18 enhances LUAD progression by targeting the miR-34a-5p/RHAMM axis, promoting tumor growth.	[69]
**circRNA (hsa_circ_0005273), miRNA** **(miR-509-3p)**	Breast Cancer (BC)	BC tissues and cell lines, mouse xenograft model	hsa_circ_0005273 boosts BC malignancy by sponging miR-509-3p to upregulate RHAMM, offering a new therapeutic target.	[70]
**miRNA** **(hsa-miR-411-5p)**	Ovarian Cancer	OVCAR-8, SKOV3 cell lines	Negatively regulates RHAMM, affecting ERK1/2 activity and cancer cell proliferation/motility.	[71]
**lncRNA** **(HMMR-AS1)**	Basal-like Breast Cancer	MDA-MB-231, MDA-MB-468 cell lines	Regulates RHAMM, affecting cancer cell proliferation and migration.	[72]
**lncRNA** **(HMMR-AS1)**	Glioblastoma	Human glioblastoma cell lines U87, U251, A172, and U118	HMMR-AS1 increases RHAMM expression and glioblastoma growth.	[73]
**lncRNA** **(HMMR-AS1)**	LUAD	LUAD tissues, mouse xenografts	HMMR-AS1 promotes tumor growth and metastasis by acting as a ceRNA for miR-138, affecting the sirt6 pathway.	[74]
**lncRNA** **(HMMR-AS1)**	HCC	HCC tissues and cells	Facilitates tumor progression by affecting macrophage polarization via miR-147a/ARID3A under hypoxia.	[75]
**lncRNA** **(HMMR-AS1)**	Glioma	Glioma cell lines (LN229, T98, and A172)	HMMR-AS1 inhibits miR-7 and upregulates CDK4 to induce glioma progression.	[76]
**lncRNA** **(HELLPAR), miRNAs** **(hsa-let-7i-5p)**	Gliomas	Glioma tissues	RHAMM shows heightened expression in gliomas, highlighting the HELLPAR-hsa-let-7i-5p-RRM2 network’s crucial role in predicting glioma outcomes.	[77]

**Table 3 ijms-26-07198-t003:** Summary of RHAMM-targeting peptides, HA oligomers, and antibodies: experimental models and observed biological effects.

Strategy	Cell/Model	Effect	Ref.
**NPI-110** **peptide**	Bleomycin-induced murine systemic sclerosis model; fibroblasts	Reduced dermal thickness, collagen accumulation, and profibrotic markers (Tgfb1, c-Myc, Col1a1, and Col3a1); increased adiponectin and perilipin expression	[91]
**NPI-106** **peptide**	Bleomycin-induced murine systemic sclerosis model; fibroblasts	Reduced profibrotic gene expression to a lesser extent than NPI-110	[91]
**P15-1** **peptide**	Rat full-thickness excisional wound model; fibroblasts	Inhibited fibroblast migration and myofibroblast differentiation; reduced TGF-β1, α-SMA, collagen I, and macrophage infiltration; promoted regenerative healing	[92]
**HA oligomers**	Fibrosarcoma HT1080 cells; various cancer cells	Reduced RHAMM-dependent adhesion and migration via suppression of ERK1/2 and FAK phosphorylation; disrupted HA–CD44–RHAMM complexes; context-dependent effects including potential transient ERK activation and wound repair responses	[97,98]
**Anti-RHAMM antibody**	Malignant B cells (hairy cell leukemia); fibroblasts; cancer cell lines	Blocked HA binding; inhibited cell motility and HA-induced locomotion; interfered with cytoskeletal reorganization; suggested potential to limit tumor invasiveness	[100]

**Table 4 ijms-26-07198-t004:** Overview of RHAMM-targeted immunotherapies in cancer research and clinical trials.

Products	Clinical Stage	Vaccine Targeting	Main Results	Ref.
**Not specified**	Not mentioned	Multiple LAAs including RHAMM	Identified LAAs including RHAMM induce humoral immune responses in leukemia patients, suggesting polyvalent vaccination could be an option for immunotherapy.	[103]
**Not specified**	Pre-clinical	LAAs including RHAMM	DCs from AML patients express LAAs like RHAMM and have the necessary HLA and co-stimulatory molecules, indicating potential for immunotherapy.	[104]
**R3 peptide** **(ILSLELMKL)**	Clinical trial initiated	RHAMM for AML	Identified R3 and R5 peptide as CD8+ T-cell epitopes in RHAMM, with AML patients’ T cells recognizing these epitopes, leading to the initiation of a clinical vaccination trial with R3 peptide.	[105]
**DCs pulsed with tumor lysates or apoptotic bodies**	Early clinical trial	B-CLL, targeting RHAMM	Demonstrated feasibility and safety of DC vaccination in CLL patients, with observed immunological responses, including an increase in specific CTLs against RHAMM.	[106]
**RHAMM-R3 peptide**	Pre-clinical/Research phase	RHAMM in CML	Demonstrated specific T-cell responses to RHAMM in CML, suggesting RHAMM-R3 peptide as a promising immunotherapy target.	[107]
**RHAMM-R3 peptide**	Post-allogeneic stem cell transplantation (SCT) research	RHAMM in CML	R3 peptide induces specific CD8+ T cell responses in CML patients post-SCT and healthy donors, with potential for augmenting graft-versus-leukemia effects.	[108]
**RHAMM mRNA-transfected DCs**	Pre-clinical	Glioma	Vaccination led to significantly longer survival and increased T lymphocyte activation, indicating strong antitumor effects in a mouse glioma model.	[109]
**Autologous DCs pulsed with B-CLL lysate**	Early clinical trial	B-CLL, targeting RHAMM among others	Vaccination resulted in increased specific CD8+ T cells against RHAMM and decreased regulatory T cells, and showed potential immunological and hematological responses in B-CLL patients.	[110]
**High-dose RHAMM-R3 peptide**	Clinical trial	AML, myelodysplastic syndromes (MDS), and multiple myeloma (MM) targeting RHAMM-R3	Immunological responses observed in 44% of patients; clinical effects seen in three patients, suggesting RHAMM-R3 as a promising target for immunotherapy, though higher doses did not increase response frequency or intensity.	[111]
**R3 peptide in incomplete Freund’s adjuvant with GM-CSF**	Phase I clinical trial	CLL targeting RHAMM-derived epitope R3	Safe vaccination; elicited specific CD8+ T-cell responses in 5 of 6 patients, with some clinical responses and induction of regulatory T cells observed.	[112]
**RHAMM-derived epitope R3**	Phase I/II clinical trial	CLL targeting RHAMM	Vaccination induced R3-specific cytotoxic T cells and profound immunological changes, suggesting effectiveness in CLL immunotherapy.	[113]
**Not applicable**	Research phase	AML targeting RHAMM	RHAMM does not differ significantly in expression between AML stem cells and healthy stem cells, reducing its suitability as a target for immunotherapy in AML.	[114]
**RHAMM mRNA-electroporated mo-DCs**	Research phase	AML targeting RHAMM	Classical mo-DCs inherently express and present RHAMM, sufficient to activate RHAMM-specific T cells without the need for electroporation. RHAMM-specific T cells were found at vaccination sites in AML patients, suggesting existing cancer immunotherapy using mo-DCs already targets RHAMM.	[115]

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
