# Peer review of "Targeting RHAMM in Cancer: Crosstalk with Non-Coding RNAs and Emerging Therapeutic Strategies Including Peptides, Oligomers, Antibodies, and Vaccines"

_ijms, 2025, doi:10.3390/ijms26157198_

Round 1

Reviewer 1 Report

Comments and Suggestions for Authors

This comprehensive review systematically summarizes the role of Receptor for Hyaluronan-Mediated Motility (RHAMM) in cancer invasion and metastasis. Notably, it highlights the regulatory mechanisms involving non-coding RNAs (ncRNAs) and evaluates various therapeutic strategies targeting RHAMM, including peptide-based therapies, oligomers, monoclonal antibodies, and vaccine development, all of which have shown promise in clinical application. While the review provides valuable contents, its organizational structure could be improved through better logical flow and more balanced distribution of content details. 

The following specific recommendations are proposed for improvement:

  1. We recommend consolidating sections 2, 3, and 4 under a unified heading structure, similar to the organization used in section 5.
  2. On line 150, the full nameof RHAMM should be inserted at its first mention.
  3. Lines 217 and 233 containrepetitive text.
  4. In Table 1, the term "median" is incorrectly formatted and should appear on the same line.
  5. Line 262 should replace “[54] [55] [56] [57] [58]” with [54-58].
  6. In Figure 4, the current depiction of CD44 in tumor progression appears inaccurate. We recommend replacing the tumor progression tissue illustration with a tumor cell representation for better clarity.
  7. References 20 and 21 are overly lengthy; author names may be omitted for conciseness.
Comments on the Quality of English Language

The manuscript would benefit from enhanced writing quality to increase its scientific rigor and readability.

Author Response

  1. We recommend consolidating sections 2, 3, and 4 under a unified heading structure, similar to the organization used in section 5.

⟶ Thank you for the helpful suggestion. Sections 2, 3, and 4 have been revised by adding unified subheadings, following the structural format used in Section 5. This change improves the clarity and coherence of the manuscript.

  1. On line 150, the full nameof RHAMM should be inserted at its first mention.

⟶ I corrected it.

  1. Lines 217 and 233 containrepetitive text.

⟶ The repetitive sentence on line 233 has been removed as suggested.

  1. In Table 1, the term "median" is incorrectly formatted and should appear on the same line.

⟶ The formatting of the term “median” in Table 1 has been corrected to appear on the same line for improved readability.

  1. Line 262 should replace “[54] [55] [56] [57] [58]” with [54-58].

⟶ The formatting issue has been corrected as suggested.

  1. In Figure 4, the current depiction of CD44 in tumor progression appears inaccurate. We recommend replacing the tumor progression tissue illustration with a tumor cell representation for better clarity.

⟶ Thank you for your valuable suggestion. Figure 4 has been revised by replacing the tissue-like illustration with a single tumor cell representation.

  1. References 20 and 21 are overly lengthy; author names may be omitted for conciseness.

⟶ Thank you for the suggestion. References 20 and 21 have been revised by abbreviating the author lists using “et al.” for improved readability.

Reviewer 2 Report

Comments and Suggestions for Authors

This review summerizes the research on RHAMM and its interaction with ncRNAs. It is releviant and of interest. The review demonstrates understanding of the current scientific knowlege and does not overstate its claims.  This was a really well written and thought out review; however there are some minor suggestions.

This review summarizes the research on RHAMM and its interaction with ncRNAs. It is relevant and of interest. The review demonstrates understanding of the current scientific knowledge and does not overstate its claims.  This was a well-written and thought-out review; however, there are some minor suggestions.
1) Section 5.1 can be consolidated to avoid repetition.
2) A comparative review of peptide, antibody, and vaccine strategies would increase the impact of this review.
3) A brief section on the limitations of RHAMM would improve the information in this review.

Author Response

1) Section 5.1 can be consolidated to avoid repetition.

⟶ Thank you for your valuable feedback. The content of Section 5.1 has been consolidated to reduce redundancy and enhance clarity while preserving the key findings.

2) A comparative review of peptide, antibody, and vaccine strategies would increase the impact of this review.

⟶ A concise comparative summary of peptide, antibody, and vaccine strategies targeting RHAMM has been added to the revised manuscript to enhance the overall impact of the review.

3) A brief section on the limitations of RHAMM would improve the information in this review.

⟶ The limitations of RHAMM as a therapeutic target have been briefly addressed in the Conclusion section.

Round 2

Reviewer 1 Report

Comments and Suggestions for Authors

Recommend acceptance for publication.